# 'Thriving with bipolar disorder': The co-design of a peer-delivered group psychoeducation program and single-arm pilot feasibility evaluation protocol

**Emma Morton[1], Andrew Kcomt**[2], **Erin E. Michalak**[3]*

**1** Department of Psychological Sciences, Monash University, Clayton, Victoria, Australia, **2** Hope + Me: Mood Disorders Association of Ontario, Toronto, Ontario, Canada, **3** Department of Psychiatry, University of British Columbia, Vancouver, British Columbia, Canada

* erin.michalak@ubc.ca

## Abstract

### Background

Self-management strategies can be used by individuals with bipolar disorder (BD) to cope with symptoms and improve quality of life (QoL). Peer-facilitated psychoeducation has potential to diversify delivery of self-management information by capitalising on the expertise of individuals who live well with BD. This protocol describes the process of co-designing a novel peer-facilitated, QoL-focused, group psychoeducation program for people living with BD, and plans for its pilot evaluation.

### Methods

Content from two web-based, self-directed psychoeducational interventions was adapted to inform a peer-facilitated group program, using a community-based participatory research (CBPR) framework. The resultant program contains eight weekly two-hour sessions on topics related to QoL in BD, and contains a combination of education, opportunities for peer-to-peer knowledge exchange, and activities that facilitate practice of self-management strategies. A single-arm pilot evaluation of this program is planned: individuals who self-identify as living with BD ($\sim n = 40$) will be recruited from the community. Four groups ($\sim n = 10$) will be delivered online by peer facilitators. The primary outcome will be feasibility (session attendance). Data will also be collected on fidelity, intervention acceptability, and impacts (QoL, mood symptoms, self-stigma, subjective recovery, self-efficacy, self-compassion, social support). A subset of participants ($\sim n = 12$) and peer facilitators ($\sim n = 4$) will be invited to participate in a feedback interview post-intervention.

**Data availability statement:** No datasets were generated or analysed during the current study. On study completion, deidentified individual participant data that underlie results reported in any published articles will be made available upon receipt of a methodologically sound, ethically approved proposal. Qualitative data will not be shared to protect privacy and confidentiality due to the potential risk of participant re-identification.

**Funding:** This research was supported by a philanthropic donation from the Laurel Foundation (https://laurelfoundation.ca/). The grant was awarded to EEM and EM. The funders had no influence on the design, conduct, or reporting of the study.

**Competing interests:** EM has received honorarium for advising on the development of unrelated educational materials for Neurotorium, an online educational platform supported by the Lundbeck Foundation. AK declared no competing interests with respect to the research, authorship, and/or publication of this article. EEM has received funding to support patient education initiatives from Otsuka-Lundbeck. This does not alter our adherence to PLOS ONE policies on sharing data and materials.

**Abbreviations:** ANOVAs, Analysis of variance; BD, Bipolar disorder; CBPR, Community-based participatory research; CREST.BD, Collaborative Research Team to Study Psychosocial Issues in Bipolar Disorder; DSM-IV, Diagnostic and Statistical Manual of Mental Disorders, fourth edition; PHQ-8, Patient Health Questionnaire-8; PHQ-9, Patient Health Questionnaire-9; QoL, Quality of life; QoL.BD, Quality of Life in Bipolar Disorder questionnaire.

## Discussion

Recovery-oriented healthcare frameworks emphasise a focus on patient-valued outcomes and the development on a peer workforce. By evaluating this novel intervention, we hope to lay the groundwork for peer-facilitated programs specific to the priorities of individuals with BD, that may be embedded in clinical settings.

## Trial registration

**ClinicalTrials.gov** NCT06878937

## Introduction

Although bipolar disorder (BD) is a serious, chronic, mental health condition, research shows that many people with this diagnosis learn to cope with their symptoms and live meaningful and satisfying lives. Individuals who live well with BD report that use of self-management strategies supports their ability to maintain mood stability and achieve good quality of life (QoL) [1–3]. Despite the fact that self-management has been elevated as a key component of all major treatment guidelines for BD, the availability of self-management education and support is limited. Only 54% of individuals receiving pharmacological treatment report accessing psychosocial services through which they may learn about and be supported to implement self-management strategies [4]. Even when available, self-management interventions often fail to holistically address patient-valued goals such as QoL and wellbeing [5,6].

Peer support (where individuals with shared lived experience of a mental health condition provide each other with informational, emotional, and social support) may be an acceptable way to disseminate information on self-management strategies, capitalising on the expertise and knowledge of people who live well with BD or related diagnoses. Indeed, up to a third of individuals with BD report turning to online or in-person peer support when seeking information related to their diagnosis [7]. Reviews of peer support for mental health challenges in general have described small positive effects on recovery-oriented outcomes, such as stigma, hope, QoL, subjective recovery, self-efficacy, and empowerment [8–10]. Although limited research has been conducted on the efficacy peer support in BD [11], there is evidence to suggest that peer support is a desired element of BD self-management programs [12], and peer contact can also encourage engagement with BD psychoeducation [13,14]. Qualitative evidence supports the acceptability of such interventions from the perspective of recipients with lived experience of BD [15–17], and provides preliminary evidence of benefits for self-stigma, hope, and self-efficacy [18–20].

A structured way to integrate peer support in the delivery of BD self-management information is through peer-facilitated group psychoeducation, which combines the benefits of evidence-based self-management information and tools, with the added advantage of providing role models for recovery and modelling of self-management skills [18,21]. A meta-analysis of 17 peer-facilitated self-management and recovery education interventions for individuals with severe and long-term mental health

conditions demonstrated small-to-medium effects for symptom severity, self-perceived recovery, hopefulness, and empowerment [22]. Only one program included in this review specifically focused on BD, however in this self-management education program peers were involved as coaches and did not deliver the psychoeducational material themselves [21]. Of the other included studies, 12 included people with BD and two included individuals with unspecified affective disorders as participants in program evaluations open to individuals of varying diagnoses, however program content was not specifically tailored to BD. One study, not included in the previous review, has evaluated a peer-facilitated self-management program in the context of BD [17]. However, the evaluation sample was small, with five participants with BD in the pilot, and five in the intervention arm. The dearth of BD-specific peer-facilitated self-management psychoeducation is a limitation, given that some challenges experienced by people with BD appear to be specific to the condition, and likely benefit from tailored information and strategies [23,24].

To capitalise on the potential of peer support to enhance the delivery of BD self-management information, this study aimed to develop a peer-facilitated, QoL-focused psychoeducation program, in partnership with people with lived experience of BD. We also aim to evaluate the feasibility, acceptability, and preliminary outcomes of this program in a single arm pilot evaluation.

## Methods

### Overview

This project extends on a program of work developed by the Collaborative Research Team to Study Psychosocial Issues in Bipolar Disorder (CREST.BD), an international research network dedicated to improving the health and wellbeing of people with BD. CREST.BD specialises in community-based participatory research (CBPR), in which researchers and knowledge users (healthcare providers, people living with BD, and their caregivers/supports) partner to design, conduct, interpret and disseminate research [25,26].

The project will be implemented across two phases. In the first phase, which has been completed, we applied CBPR principles to develop a peer-facilitated, QoL-focused group psychoeducation program for individuals with BD. In the second phase, we will conduct a pilot evaluation of the feasibility, acceptability, and preliminary outcomes of the program.

### Phase 1. Development of the 'Thriving with bipolar disorder' program

**Theoretical framework.** Self-management describes the behaviors implemented by an individual to monitor and respond to the symptoms and impacts of a chronic illness [27,28]. The focus of self-management extends beyond symptom management alone, and incorporates strategies to support physical, social, and psychological aspects of QoL [29]. Evidence-supported self-management strategies for BD include: medication adherence, mood monitoring, sleep and lifestyle regularity, exercise, healthy diet, detecting and responding to early signs of relapse, and stress-management [1–3,30,31].

The 'Thriving with Bipolar Disorder' program was adapted from psychoeducational content developed for two previous CREST.BD web-based, self-directed self-management interventions: The Bipolar Wellness Centre website and the PolarUs app [19,32]. In both interventions, information and resources to support self-management are organised according to a theoretical conceptualisation of QoL as it is specifically impacted in BD [24]. This model of QoL incorporates life areas directly affected by BD symptoms and medications (mood, sleep, physical health, cognition), functioning and participation (household management, leisure, finances, relationships), and subjective experiences (self-esteem, spirituality, identity, independence). Two optional areas, work and study, can be included depending on the roles an individual is engaged in. The factor structure of the 12 core QoL domains was supported through psychometric evaluations of the Quality of Life in Bipolar Disorder scale (QoL.BD), a BD-specific QoL self-assessment which operationalises and measures QoL according to this framework [24,33].

The rationale for integrating peer support with content adapted from these self-directed programs was derived from both the theoretical literature and participant feedback. Social learning theory posits that peers can act as role models for effective self-management [34], thereby increasing participants' sense of self-efficacy [35]. Similarly, social comparison theory suggests that individuals benefit from seeing peers who live well despite similar challenges, as this can inspire hope for successful outcomes of behavior change [36], as well as reduce self-stigma, which can act as a major barrier to help-seeking [9]. Participant feedback in the evaluation of the Bipolar Wellness Centre aligns with these theoretical justifications [19], highlighting that participation in accompanying knowledge translation strategies with a greater degree of peer contact (face-to-face group workshops, one to one online peer 'Living Libraries') resulted in greater self-efficacy improvements than strategies without extensive peer contact (online videos and webinars). Qualitative evidence from this evaluation indicated that peer contact can normalise BD-related challenges, promote greater self-compassion towards one's circumstances, enhance understanding of effective self-management strategies, and encourage a more optimistic outlook on the possibility of living well with BD [19,37].

**Community-based participatory research framework.** A CBPR framework was used to guide development of the 'Thriving with Bipolar Disorder' program and content. CREST.BD specialises in a CBPR model that considers the unique strengths and challenges of people living with BD [25,26], fostering authentic collaboration and partnership between academics, healthcare providers, caregivers, and people with lived experience. CBPR methods were used to develop the QoL framework and psychoeducation materials that informed 'Thriving with Bipolar Disorder' content: the QoL.BD was developed through extensive consultation with people with lived experience [23], and content and resources for the Bipolar Wellness Centre and PolarUs app were co-authored and curated via a collaborative process [19,32]. The 'Thriving with Bipolar Disorder' program therefore leverages over a decade of CBPR-based efforts to understand and improve QoL-focused self-management in QoL.

To incorporate the perspectives of individuals with lived experience of BD in the 'Thriving with Bipolar Disorder' program and content, two existing CREST.BD advisory groups were engaged. One advisory group has a longstanding history of consulting at a high-level on the network's program of QoL-focused research and knowledge translation. The other advisory group consulted specifically on the development of the PolarUs app for BD [32], and in many instances had been involved in co-authoring app content. Advisory groups were consulted over seven virtual feedback sessions between November 2023 and July 2024 to inform the content and delivery of the peer-facilitated program, including program and session length, session format (i.e., adaptations for online vs in person delivery), and balance of didactic information delivery to interactive exercises and peer discussion. Group members participated in prioritisation exercises to select content areas and specific self-management strategies for inclusion.

The advisory groups were also involved in developing and piloting group activities. Advisory group member responses to these pilot activities were incorporated as resources for peer facilitators in order to demonstrate activity completion. These resources are also intended to be utilised during program delivery to support group members in brainstorming activity responses; for example, advisory group responses can be used as a 'card sorting' activity to help participants identify and categorise early warning signs of mood episodes.

**Program structure and content.** Following the CBPR consultation activities described above, two program manuals were drafted by EM and reviewed by EEM, adapting and extending on content developed for the Bipolar Wellness Centre and PolarUs app [19,32]. The facilitator manual contains detailed instructions on content to be reviewed and exercises to be conducted for each session, including activity timing and prompts to facilitate group discussions. The attendee manual summarises psychoeducational content and self-management strategies discussed in session and provides space to record personal reflections.

The program consists of eight weekly two-hour sessions that will be co-facilitated by two peer facilitators with lived experience of BD or a mood disorder. Each session follows the same format: peer facilitators will present an overview of the session, before facilitating a brief review of participants' experiences of the previous week's take-home activities.

Peer facilitators will then provide psychoeducation on the week's QoL topic and relevant self-management strategies, and facilitate group discussions to reinforce specific concepts and provide opportunities for peer support and learning. They will also guide completion of structured skills practice activities, designed to support participants to understand and apply the self-management strategies. A session-by-session overview of topics, psychoeducational content, and exercises is provided in Table 1.

**Phase 2. Pilot evaluation of the 'Thriving with bipolar disorder' group program**

**Study design.** The 'Thriving with bipolar disorder' evaluation will be a single-arm pilot feasibility trial. An explanatory sequential mixed-methods design will be used [38]; semi-structured qualitative interviews will be conducted with a subset of pilot participants to deepen understanding of quantitative results. The study has been registered with ClinicalTrial.gov (NCT06878937). The SPIRIT (Standardized Protocol Items: Recommendations for Interventional Trials) checklist has been followed (S1 File).

Four groups will participate in the pilot. Recruitment of facilitators commenced June, 2025, and is expected to conclude by July, 2025. Recruitment of program participants began July, 2025 and concluded August, 2025. Data collection is expected to occur between August, 2025 – January, 2026. Results are expected by June, 2026.

Table 1. Overview of 'Thriving with Bipolar Disorder' session content.

| Session topic | Session objectives |
|---|---|
| 1. Introduction | Establish a warm supportive environment, set expectations for the program, collaboratively establish group guidelines, discuss safety.<br>Review background concepts (self-management, quality of life).<br>Demonstrate how to complete a quality of life self-assessment. |
| 2. Mood | Establish the importance of detecting and responding to early warning signs of mood episodes.<br>Describe the key components of an early intervention plan.<br>Demonstrate how to monitor daily mood fluctuations. |
| 3. Sleep | Establish the importance of sleep for mood stability.<br>Describe the three factors supporting good sleep.<br>Highlight the importance of regular routines.<br>Demonstrate use of the activity planner. |
| 4. Physical health | Describe the impacts of bipolar disorder symptoms and medications on physical health.<br>Describe the impacts of diet and exercise on mood and quality of life.<br>Discuss barriers to making lifestyle changes and potential solutions.<br>Review the use of the mood monitoring and daily planner worksheets, and their relevance for making lifestyle changes. |
| 5. Relationships | Establish the importance of social connections and social support.<br>Describe the different types of social connections and social supports.<br>Discuss potential benefits and risks of disclosing a bipolar diagnosis.<br>Introduce a structured method to plan a disclosure strategy. |
| 6. Money | Describe the relationship between bipolar disorder symptoms and money difficulties.<br>Discuss factors that increase the risk of impulsive spending.<br>Introduce barriers to spending and slow down and delay strategies.<br>Highlight free and cheap ways to have fun. |
| 7. Self-esteem | Clarify the difference between self-esteem and self-compassion.<br>Identify impacts of stigma.<br>Practice ways to offer self-compassion to ourselves.<br>Learn methods to self-soothe using the five senses. |
| 8. Independence | Establish the importance of independence.<br>Review the difference between values and goals.<br>Highlight the accomplishments and strengths of group attendees.<br>Conclude the program on a positive note. |

 

**Ethics.** Program facilitators and participants will receive written information about the study and indicate their informed consent before proceeding; participants will be invited to formally reconsent to the study prior to completing the post-intervention (8 week) and follow-up (12 week) surveys. Data will be treated confidentially and stored on a secure server in Canada. The study has received ethical approval from the University of British Columbia Behavioural Research Ethics board (H24-03489). A copy of the ethics-approved protocol, including risk and data management processes, can be viewed in S2 File.

**Study setting.** The 'Thriving with Bipolar Disorder' program will be delivered through Hope+Me, a charity based in Ontario, Canada, that supports people living with mood and anxiety disorders through education, advocacy, training, and support services. Hope+Me has one of the largest community infrastructures within the national network of mood disorders associations, serving over 76,000 people province-wide. Hope+Me also provides services tailored to the needs of marginalised and racialised communities, such as dedicated LGBTQ+ and Black, Indigenous, and People of Colour peer support groups.

**Peer facilitator recruitment and training.** Four peer facilitators will be recruited through Hope+Me. Inclusion criteria for facilitators are: [1] aged 18 or older, [2] self-reported diagnosis of a mood disorder, [3] access to an internet-enabled computer or smartphone device through which they can access the Zoom teleconferencing platform, [4] ability to understand, read, and write English to a sufficient level to allow participation in the program and research activities, [5] completion of Hope+Me peer support training and at least one year of prior experience leading peer support groups or recovery programs through Hope+Me, and [6] completion of a Criminal Records Check. Peer facilitators will receive an honorarium of $50 CAD per hour for their participation, including time associated with training, program delivery, and data collection.

Hope+Me facilitators will have already received a minimum of 300 hours of training in peer support, de-escalation techniques, suicide risk management (safeTALK), and supporting personal recovery (Wellness Recovery Action Plan training). Facilitators also shadow a trained facilitator when first starting. Specific training will be provided on the 'Thriving with Bipolar Disorder' program prior to intervention delivery: facilitators will attend four hours of training provided by co-author EEM. This training will review the program structure, delivery, session-by-session objectives, and use of the facilitator manual. Training will also cover aspects of the research process that relate to the facilitator role, such as consent, maintaining privacy and confidentiality and managing conflicts of interest.

Ongoing supervision will be provided to the facilitators via existing Hope+Me channels, which include peer debriefing with co-facilitators following sessions. Peer facilitators also have access to support from Hope+Me counsellors and/or staff members during and following sessions. Lastly, peer facilitators have access to a monthly community of practice which includes discussion of strategies to manage challenging situations, such as recognising and addressing conflicts in group settings.

**Program participants and recruitment.** Program participant recruitment will occur via promotion via the Hope+Me newsletter, social media channels, and existing peer support programs, as well as notices on CREST.BD social media pages, paid advertisements on Facebook, Instagram, and X/Twitter, emails to the CREST.BD mailing list, and Canadian healthcare providers or organisations associated with the CREST.BD network.

Inclusion criteria for program participants are: (1) aged 18 or older, (2) residing in Canada, (3) self-reported BD diagnosis (Type I, II, or Not Otherwise Specified), (4) ability to read, understand, and write English to a sufficient level to allow participation in the program and research activities, and (5) sufficient access to an internet-enabled computer or smartphone device through which they can access the Zoom teleconferencing platform. Participants will be permitted to continue their usual treatments (pharmacological and/or psychological) during the program.

Participants will be reimbursed with a $30 (CAD) gift card for completing each evaluation survey during the program (for a maximum of $90 per participant). Individuals who participate in the optional follow-up qualitative interview will be provided with an additional $30 (CAD) gift card. Participants who choose to withdraw partway through the study will still be offered the honorarium for study activities they participated in.

We will aim to recruit 32–40 participants across four groups (8–10 per group) given previous research summarising the ideal group size [39]. As the main aim of this trial was to assess feasibility and acceptability, no formal power calculations were conducted. However, samples above 30 are generally considered acceptable for pilot studies [40].

**Procedure.** The 'Thriving with bipolar disorder' program will be conducted over the Zoom teleconferencing platform. The structure and content of the program is detailed above.

Facilitator and participant recruitment will begin in April, 2025. Interested facilitators will be sent a link to the online survey platform, Qualtrics, where they can review the explanatory statement, register their consent to participate, confirm their eligibility, and provide demographic and clinical information. Programs will not commence until consenting facilitators have undergone training in the 'Thriving with Bipolar Disorder' intervention (described above). Peer facilitators will complete a brief post-session Qualtrics survey each week, recording attendance, fidelity, and session feedback.

Interested participants will be provided a Qualtrics link, where they will be able to review the explanatory statement, register their consent to participate, confirm their eligibility and availability, and provide baseline demographic and clinical information.

As this is a non-randomised trial, allocation to program groups will be based on convenience. Eligible, consenting participants will be able to self-select into a program timeslot until group capacity is met ($n = 10$). Participants will be sent calendar invites containing the Zoom links for each session, and instructions on how to download and use Zoom.

Prior to the first session, participants will be sent a link to a Qualtrics form where they will complete baseline outcome measures. Following this, they will be provided with a copy of the attendee manual. As 'Thriving with Bipolar Disorder' was designed as a closed program (that is, no new members are added after the group begins), individuals who do not attend the first session will be treated as withdrawn.

Immediately post intervention completion, participants will be provided with a link to a Qualtrics form to provide program feedback and complete outcome measures. Participants will be re-contacted after 1 month to complete final, follow-up outcome measures.

At the end of the intervention period, a subset of consenting participants ($\sim n = 12$) and peer facilitators ($\sim n = 4$) will be invited to participate in a one-hour qualitative individual interview. All interviews will be conducted remotely via Zoom and will be recorded and transcribed for later analysis.

**Data collection.** Both quantitative and qualitative data will be collected to characterise feasibility, acceptability, and relevant clinical, QoL, and recovery-oriented outcomes. The primary outcome will be feasibility (session attendance). The type and timing of data collection is summarised in Fig 1.

*Demographics:* At enrollment, facilitators and program participants will be asked to provide information on demographics (age, gender, cultural and racial background, education, occupation) and self-reported clinical characteristics (BD diagnosis, age of symptom onset and diagnosis, number of mood episodes, current treatment).

*Feasibility:* Feasibility will be assessed using program attendance rates and facilitator-reported fidelity.

After each session, peer facilitators will complete a brief questionnaire to record the number of participants who attended each session, self-report fidelity to the intervention as described in the manual, and provide feedback on the suitability of the intervention. Feasibility was defined as the proportion of participants achieving an attendance of 62.5% (i.e., five out of eight sessions), based on the median therapeutic dose in a review of group therapy program evaluations [39]. Reasons for non-attendance will be recorded by facilitators using open text items in the weekly feedback questionnaire.

To evaluate fidelity, facilitators will self-report adherence to the manual using four self-report Likert-type items, asking them to report the degree to which they reviewed information as described in the manual, shared their own lived experiences, and facilitated group involvement in discussions and activities. Fidelity will be defined as a rating of 4–5 ('agree' or 'strongly agree') across all items. This method of fidelity assessment was selected given that the use of external observers may impact participant willingness to participate in group discussions and activities. Similar self-reported fidelity

| | Study Period | | | | | | | | | | | |
| --- | --- | --- | --- | --- | --- | --- | --- | --- | --- | --- | --- | --- |
| | Enrollment | Allocation | Post-allocation | | | | | | | | | |
| **Timepoint (Week)** | -t | 0 | 1 | 2 | 3 | 4 | 5 | 6 | 7 | 8 | 12 |
| **Enrollment** | | | | | | | | | | | | |
| Eligibility screen | X | | | | | | | | | | | |
| Informed consent | X | | | | | | | | | | | |
| Allocation | | | | | | | | | | | | |
| **Interventions** | | X | | | | | | | | | | |
| Delivery of 'Thriving with Bipolar Disorder' | | | X | X | X | X | X | X | X | X | | |
| **Assessments** | | | | | | | | | | | | |
| Baseline sociodemographic and clinical characteristics | X | | | | | | | | | | | |
| *Feasibility:* | | | | | | | | | | | | |
| Session attendance *(Facilitators)* | | | X | X | X | X | X | X | X | X | | |
| Self-reported program adherence *(Facilitators)* | | | X | X | X | X | X | X | X | X | | |
| *Acceptability:* | | | | | | | | | | | | |
| Program and facilitator feedback | | | | | | | | | | | X | |
| Client Satisfaction Questionnaire-8 | | | | | | | | | | | X | |
| Session feedback *(Facilitators)* | | | X | X | X | X | X | X | X | X | | |
| Group Climate Questionnaire Short Form *(Facilitators)* | | | X | X | X | X | X | X | X | X | | |
| *Outcomes:* | | | | | | | | | | | | |
| Quality of Life in Bipolar Disorder Scale *(Participants)* | | X | | | | | | | | | X | X |
| Patient Health Questionnaire 8 *(Participants)* | | X | | | | | | | | | X | X |
| Altman Self-Rating Mania Scale *(Participants)* | | X | | | | | | | | | X | X |
| Internalized Stigma of Mental Illness Scale *(Participants)* | | X | | | | | | | | | X | X |
| Bipolar Recovery Questionnaire *(Participants)* | | X | | | | | | | | | X | X |
| Stanford's Chronic Disease Self-Efficacy 'Manage Disease in General' subscale *(Participants)* | | X | | | | | | | | | X | X |
| Self-compassion scale Short-Form *(Participants)* | | X | | | | | | | | | X | X |
| Social Provisions Scale-Short Form *(Participants)* | | X | | | | | | | | | X | X |
| *Qualitative feedback:* | | | | | | | | | | | | |
| Individual interviews *(Participants, facilitators)* | | | | | | | | | | | X | |

**Fig 1. SPIRIT (Standard Protocol Items: Recommendations for Clinical Trials) schedule of enrollment, intervention, and assessments.**

assessments have previously been used in evaluations of peer-led programs [41,42] and peer support services [43]. Reasons for non-fidelity will be explored in qualitative interviews with facilitators (see below).

*Acceptability:* The Client Satisfaction Questionnaire-8 [44] will be used to assess attendees' satisfaction with the 'Thriving with Bipolar Disorder' program post-intervention. This instrument has high internal consistency (Cronbach $\alpha = 0.91$); higher scores have been associated with greater attendance of services and improved clinical outcomes [45].

We will query attendee perceptions of program content, delivery, the facilitator, and the group environment, using a series of Likert-scale statements developed by the research team (1 = "Strongly Disagree" to 5 = "Strongly Agree"). Acceptability will be defined as a rating of 4–5 ('agree' or 'strongly agree') across all items. Reasons for non-acceptability will be explored in qualitative interviews with attendees (see below).

To describe the acceptability of the group climate, we will use the Group Climate Questionnaire short form [46]. This is the most widely used measure of group therapy cohesion [47], and can be completed by a group member, facilitator, or observer [48]; here, to minimise response burden on attendees, weekly feedback will be provided by facilitators. It contains 12 items describing specific behaviours that are rated on a 7-point Likert scale (where 0 indicates "not at all" and 7 indicates "extremely"), three subscales can be calculated based on the mean score of relevant items. These subscales are Engaged (which describes a positive working group atmosphere), Avoiding (which describes avoidance of responsibility for group dynamics), and Conflict (which describes tension and anger between group members).

***Program outcomes:*** To evaluate program outcomes, participants will complete a set of measures using Qualtrics within a week prior to the intervention commencing, immediately post-program (eight weeks), and one month following program completion (12 weeks). The outcome measures were chosen in line with the theoretical underpinnings of the intervention (described above), and include self-reported QoL, mood symptoms, personal recovery, self-efficacy, self-compassion, self-stigma, and social support.

The Quality of Life in Bipolar Disorder (QoL.BD) Scale will be used to assess aspects of QoL specifically impacted by or relevant to BD [24]. This scale assesses 12 core aspects of QoL (mood, sleep, physical health, cognition, household management, leisure, finances, relationships, self-esteem, spirituality, identity, and independence). Each domain contains four Likert-type items that assess satisfaction with that area (1 = Strongly Disagree to 5 = Strongly Agree). An overall score can be calculated by summing responses to the 48 items (range: 48–240), with higher scores indicating greater satisfaction with life. During psychometric testing, the factor structure of the QoL.BD has been supported [33], and the scale has demonstrated excellent internal reliability (Cronbach $\alpha > 0.8$) and appropriate test-retest reliability [24]. Construct validity has been supported through positive correlations with generic QoL instruments and negative correlations with depressive symptoms [24]. In multiple clinical trials, this instrument has demonstrated sensitivity to treatment effects [49].

Self-reported depressive symptoms will be assessed by the Patient Health Questionnaire-8 (PHQ-8) [50], which evaluates eight of the nine criteria for depression from the Diagnostic and Statistical Manual of Mental Disorders, fourth edition (DSM-IV). This instrument has demonstrated comparable performance to the original nine item version (PHQ-9), which assesses all DSM-IV criteria, including current suicidal ideation. However, this item may be inappropriate to use in settings where formal psychiatric evaluation and support is not available, or circumstances where data is self-reported and immediate follow-up is not possible [50]. Given that the 'Thriving with Bipolar Disorder' program is peer facilitated and primarily aims to provide support with QoL self-management, and data will be self-reported using Qualtrics, we considered the PHQ-8 (which omits this item) to be an appropriate and safe way to assess depressive symptoms. The eight Likert-type items ask about the frequency of depressive symptoms experienced during the past two weeks (0 = "Not at all" to 3 = "Nearly every day"). Responses are summed to create a total score (range: 0–24); higher scores indicate greater depression severity. The PHQ-8 performs comparably with clinician assessments [51,52]. Internal reliability has been demonstrated in population studies (Cronbach $\alpha = 0.87$) [53].

Self-reported mania will be assessed using the Altman Self-Rating Mania Scale [54]. This instrument measures behaviours and feelings during the past week using five items; a total score (range: 5–25) can be calculated by summing responses, with higher scores indicating more severe manic symptoms. Responses have been found to correlate strongly with clinician-rated mania [55]; internal reliability is acceptable (Cronbach $\alpha > 0.65$) [54].

Personal recovery will be measured using the Bipolar Recovery Questionnaire. Items are informed by qualitative interviews regarding experiences of personal (as opposed to clinical) recovery in BD [56]. This scale contains 36 visual analogue scales (range: 0–100); responses are summed to create an overall score (range: 0–3600). Higher scores are indicative of better self-appraised recovery. This instrument has been found to be internally consistent (Cronbach $\alpha = 0.875$) and reliable over a month-long test-retest period. It demonstrated sensitivity to change in an evaluation of a psychological intervention for BD [57].

Self-efficacy will be measured using the 5-item Manage Disease in General subscale of Stanford's Chronic Disease Self-Efficacy Scale [58,59]. The score on this subscale is the mean of 5 Likert-type items (1 = not at all confident to 10 = totally confident). Higher scores on this subscale (range: 1–10) indicate greater confidence in managing the symptoms and impacts of a chronic health condition.

Self-compassion will be measured using the Self-Compassion Scale-Short Form [60,61], which contains 12 Likert-type items (1 = Almost Never, 5 = Almost Always) used to assess 6 dimensions of self-compassion: self-kindness, self-judgement, common humanity, isolation, mindfulness, and overidentification. Subscales are calculated by averaging relevant items. To calculate an overall score (range: 1–5), negative subscales are reverse-coded. Internal consistency has been demonstrated

(Cronbach α = 0.86); confirmatory factor analysis supports both the 6 subscales and a single higher-order factor of self-compassion [61]. Higher overall scores indicate more frequent self-compassionate behaviours and attitudes.

Self-stigma will be measured using the Internalised Stigma of Mental Illness Scale Brief Version [62]. This instrument contains 9 Likert-type items (1 = Strongly Disagree, 4 = Strongly Agree); two are reverse-coded. Item responses are averaged for an overall score (range: 1–4); higher scores indicate more internalised stigmatising attitudes towards mental illness. Internal consistency has been demonstrated (Cronbach α = 0.86); confirmatory factor analysis supports a unidimensional factor structure [63].

Perceived social support will be measured using the Social Provisions Scale Short Form [64]. This instrument contains 10 Likert-type items (1 = Strongly Disagree, 4 = Strongly Agree) which evaluate an individual's perceptions of the availability of social support. An overall score (range: 10–40) and five subscales can be calculated by summing relevant items; subscales include guidance (receipt of information or advice), reliable alliance (practical help and support), reassurance of worth (feeling valued by others), attachment (emotional bonds), and social integration (sense of belonging to a group with shared interests and attitudes). Internal consistency of the scale has been supported (Cronbach α > 0.80) [64].

***Qualitative interviews:*** Individual interviews (approximately one hour) will be conducted with a sample of participants (~$n$ = 12) and facilitators (~$n$ = 4) post-intervention. During the post-program follow-up, attendees will be asked to express their interest in participating in an optional interview. Where possible, we will extend invitations to maximise the diversity of the interviewed sample (i.e., age, gender, BD subtype), however given the limited pool of attendees (~$n$ = 40) it is expected that the recruitment will largely be informed by convenience sampling.

A semi-structured interview guide will contain questions about the acceptability of program content and format, experiences of the group environment, perceptions of the facilitator, and changes experienced as a result of participation. Participants will also be asked for their perspectives on aspects of the program that should be modified or added. Open-ended questions will be used where possible to allow participants to focus their responses on experiences that are the most meaningful and salient from their perspective. Follow-up probes will be used to elicit depth and clarification of responses. Copies of the semi-structured interview guides for program attendees and facilitators have been provided in S3 File.

As saturation is not a recommended approach for reflexive thematic analysis [65], the initial target sample size for the qualitative interviews will be guided by both information power [66] and feasibility considerations (i.e., the number of facilitators interviewed will be constrained by the number of trained facilitators, and the number of attendees interviewed will be constrained by the number of participants who opt-in to being contacted). Aspects of information power that support the use of a smaller initial target sample were the narrowness of the study aim (that is, to evaluate the feasibility, acceptability, and preliminary outcomes of this program), specificity of the sample (program delivery will be manualised), and expected quality of the dialogue (EM is an experienced qualitative interviewer with expertise in peer support and BD self-management interventions). Information power is intended to be a flexible approach to sampling, and as such the appropriateness of the attendee qualitative interview sample size will be iteratively reviewed as data collection progresses. The richness and relevance of the interview dialogue will be discussed between research team members to inform the final sample size.

**Data analysis.** ***Quantitative data analysis:*** Quantitative data will be analysed using SPSS Version 29. Descriptive statistics will be used to summarise demographics and participant flow. As per recommendations for pilot and feasibility studies, descriptive statistics will be used to characterise feasibility and acceptability [40,67]. Responses will be reported as frequencies and percentages, or as means and standard deviations as applicable. We will report overall participant attendance rates and number of participants achieving the minimum therapeutic dose (5/8 sessions). Reasons for non-attendance will be summarised narratively. We will report overall and per session intervention fidelity. Reasons for non-fidelity will be explored in the qualitative analysis, described below. We will report attendee perceptions of intervention acceptability (per item, and an average overall acceptability score) and facilitator impressions of group climate (overall, and per session). Acceptability for facilitators and attendees will be further explored in the qualitative analysis, described below.

A secondary objective of this study is to conduct an exploratory analysis of program outcomes, noting that the pilot is not powered to detect significant differences and any insights gleaned will be preliminary in nature. Change in program outcomes will be evaluated statistically using linear mixed models, given their ability to handle missing data [68,69]. Program outcomes will be assessed for outliers and distribution normality. Separate models will be created for each outcome measure. Following the model building procedure described by Hox [68], we will assess 1) a null model, 2) a model containing fixed effects for Time (baseline, post-program and one-month follow-up), and if indicated by visual inspection, a quadratic effect, 3) a model containing random effects for participant ID, and 4) a model containing group IDs as fixed effects, following recommendations for small sample sizes in three level linear mixed models [70]. Restricted maximum likelihood estimates will be used given the small sample size [71]. Statistical significance will be set at the $p < 0.05$ level, with two-tailed analysis. Throughout, 95% confidence intervals will be presented. Following recommendations for exploratory analyses, we will not correct for multiple testing [72].

Analyses will be per protocol (including only participants meeting the feasibility threshold of attendance at 5/8 sessions) and modified intention to treat (using all data available from participants who commenced treatment) [73]. To reduce the likelihood of missing data, participants will not be able to skip items on Qualtrics. Number of missed assessments per time point will be reported. Chi-square and t-tests will be used as appropriate to evaluate the relationship between any missing data and observed values. Participants with no outcome data generated by the study (i.e., missing both baseline, post-program, and follow-up assessment data) will be excluded from the analysis.

***Qualitative data analysis:*** Qualitative data analysis will be performed using Braun and Clark's guidelines for reflexive thematic analysis [74] using NVivo 14 (QSR International, 2023). Data familiarisation will occur through reading and re-reading transcripts. Inductive coding approach will be used to assign brief descriptive codes derived from the data. Themes will be generated to describe patterns in coding. Provisional themes will be reviewed during development for coherency, meaningfulness, richness, and relationships to other themes. Results will be presented as a narrative summary of the most salient and relevant themes with respect to the study aims.

In line with quality practice recommendations for reflexive thematic analysis, a single researcher will perform the analysis [75]. The primary analyst will maintain a reflexive journal to document the potential influence of any preexisting knowledge, assumptions, or personal characteristics on the data collection and analysis process. To support rigor, credibility will be established via peer debriefing between the co-authors EM and EEM to reflect on key decisions during data collection and analysis [76]. Credibility will also be upheld through use of member checking, which will be accomplished by inviting CREST.BD advisory group members to review and comment on the definition, relevance, and interpretation of themes developed [77]. Confirmability of the results will be supported through the use of verbatim participant quotes.

***Integration of quantitative and qualitative findings:*** Qualitative and quantitative data will be analysed and reported separately; integration of will occur at the level of interpretation in a staged approach [78]. Findings from the qualitative and quantitative analyses will be compared and described in the report of qualitative findings to deepen understanding of changes observed in the quantitative analysis. Identification of any discrepancies will prompt investigation of potential reasons for conflicting results (e.g., statistical/information power, survey and interview item content).

In line with the CBPR approach described above [25,26], CREST.BD lived experience advisory groups will also be involved in the interpretation of qualitative and quantitative data. Both groups will be presented with a summary of findings and asked for their input regarding key results to highlight in any dissemination activities (i.e., publications, plain language blog summaries), insights regarding discrepancies between quantitative and qualitative findings, and suggestions for refining the intervention and evaluation methods for future trials.

## Discussion

Group psychoeducation with interactive skills practice is an evidence-supported intervention for BD [79]. However, existing psychoeducation programs are primarily focused on symptom management, despite the fact that people with BD desire

self-management information that addresses of QoL domains such as physical health, social and occupational functioning, and self-esteem [5]. Such diverse topics are commonly discussed in peer support services [21,80–83], suggesting that this context is ideal for the delivery of psychoeducation that encompasses both symptoms and QoL. The novel intervention described here, 'Thriving with Bipolar Disorder,' aims to leverage the potential of peer support for QoL-focused self-management psychoeducation in BD. The forthcoming mixed-methods evaluation study will provide useful preliminary data on the feasibility, acceptability, and outcomes of this program. Such insights will be used to further refine the program content, delivery, training materials, and evaluation methods for future trials.

## Limitations

The findings of the evaluation study must be interpreted in light of its limitations: as a small, pilot, non-randomised study, it is not powered to detect treatment effects. Furthermore, participants in the trial will be self-selected, and therefore may be more highly motivated to seek this form of support. Prior positive attitudes towards peer support may impact feasibility, acceptability, and effectiveness outcomes, and as such findings may not generalise to evaluations in other contexts. In addition, the facilitator qualitative interview sample size will be constrained by the number of facilitators trained in the intervention ($n = 4$). Although the goal of qualitative research is not generalisability [84], a small interview sample may limit the depth and diversity of facilitator experiences explored. However, we note that such pragmatic considerations are an accepted justification for constraining sample sizes in published qualitative research [85].

To reduce participant burden, we do not plan to investigate all outcomes that may be impacted by program participation. For example, given that sessions focus on a broad range of QoL topics including nutrition, exercise, sleep, and relationships, future evaluations may wish to explore consequent changes in physical health, sleep quality, and social support. To help identify candidate outcomes for larger trials, the pilot evaluation will use qualitative interviews to explore the experiences of a subset of participants, which may assist in identifying salient QoL changes for inclusion in future quantitative research.

Finally, self-assessed fidelity will be used to evaluate the extent to which the 'Thriving with Bipolar Disorder' program was delivered as intended. This method of evaluation was selected for two reasons: first, some previous studies of peer-led programs have employed self-reported fidelity assessments given the low acceptability of recordings to peer facilitators [41–43]. Second, although use of independent raters is considered the gold standard of fidelity assessment [86], this requires a significant investment in terms of cost, time, and staffing [87,88], and this pilot was not sufficiently resourced to train and retain raters. Despite these pragmatic considerations, self-reported fidelity may result in biased estimates of adherence to intervention delivery and facilitator competence given the potential for social desirability to influence responding [89,90]. Given these concerns, we intend to explore the suitability of various fidelity measures in consultation with the CREST.BD lived experience advisory groups as part of post-trial consultation activities to refine the intervention and evaluation methods [25,26]. We will discuss the possible use of observation and/or recording, as well as strategies to make such methods more acceptable. For example, peer facilitators themselves have been successfully trained to evaluate the fidelity of peer-delivered group cognitive-behavioral therapy for postpartum depression, demonstrating excellent interrater reliability with a psychiatrist and graduate student [91]. Other non-obtrusive methods to assess fidelity may also be considered; for example, previous research has reviewed worksheets completed in session of as a proxy for intervention adherence [92]. The precise method of fidelity assessment to be used in future trials will depend on a combination of participant/facilitator feedback, advisory group input, the preferences of any partner organisations involved, and the resources available.

## Conclusion

As part of the adoption of recovery-oriented frameworks for mental health service delivery, peer support programs have been embedded in healthcare settings internationally [93–95]. Through this pilot evaluation of the 'Thriving with Bipolar

Disorder' program, we hope to lay the groundwork for peer-facilitated programs specific to the needs of individuals with BD that may be embedded in clinical settings. We also see potential for this program be delivered via community or peer-led organisations: given peer support in such settings tends to be informal (i.e., activities and topics of discussion are not prescribed), peer-delivered psychoeducation could diversify support options, and may be particularly appealing to individuals seeking a more structured format [16]. Should this intervention demonstrate feasibility, acceptability, and potential for impacting recovery-oriented outcomes, this will support future evaluation in a full-scale randomised control trial, and across diverse mental healthcare contexts.

## Supporting information

**S1 File. SPIRIT Checklist 2025.** SPIRIT (Standardized Protocol Items: Recommendations for Interventional Trials) 2025 checklist of items to address in a randomized trial protocol.
(DOCX)

**S2 File. Ethics Approved Protocol.** The full protocol and ethics approval from the University of British Columbia Behavioural Research Ethics board (H24-03489), including risk and data management processes.
(PDF)

**S3 File. Semi-structured Interview Guides.** Semi-structured qualitative interview guides to be used with attendees and facilitators.
(DOCX)

## Acknowledgments

The authors gratefully acknowledge the expertise and contributions of the CREST.BD Community Advisory Groups, CREST.BD network members, and Hope+Me in the development of the 'Thriving with Bipolar Disorder' intervention and pilot feasibility study.

## Author contributions

**Conceptualization:** Emma Morton, Erin E. Michalak.

**Funding acquisition:** Emma Morton, Erin E. Michalak.

**Project administration:** Emma Morton, Andrew Kcomt, Erin E. Michalak.

**Resources:** Emma Morton, Erin E. Michalak.

**Writing – original draft:** Emma Morton.

**Writing – review & editing:** Andrew Kcomt, Erin E. Michalak.

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
