## [Decision Letter · Decision Letter 0]

1 Oct 2025

Dear Dr. Michalak,

Thank you for submitting your manuscript to PLOS ONE. After careful consideration, we feel that it has merit but does not fully meet PLOS ONE’s publication criteria as it currently stands. Therefore, we invite you to submit a revised version of the manuscript that addresses the points raised during the review process.

We look forward to receiving your revised manuscript.

Kind regards,

Rakesh Karmacharya, MD, PhD

Academic Editor

PLOS ONE

Journal Requirements:

“EM has received honorarium for advising on the development of unrelated educational materials for Neurotorium, an online educational platform supported by the Lundbeck Foundation. AK declared no competing interests with respect to the research, authorship, and/or publication of this article. EEM has received funding to support patient education initiatives from Otsuka-Lundbeck.”

**Additional Editor comments:**

Review #3: 

Overall:

Project has strong rationale, and the paper is well written.

Introduction:

Recommend rewording the first sentence to say “BD is a serious, potentially chronic mental health condition” – as symptoms are significant, no matter their duration.

The authors cite a meta-analysis of 17 peer-facilitated self-management groups for those with smi – could they specify how many of those participants across studies these studies had a diagnosis of BD? This would help establish that the uniqueness of the present study.

Methods:

For the qualitative analyses, it is unlikely saturation will be reached if only interviewing four facilitators; consider increasing this sample if possible.

Lots of detail is provided about the quantitative measures that will be given, but the qualitative interview guide is not as specific. Consider including questions that will be asked to parallel the specificity of quantitative measures described.

The authors state that participants will not be able to skip questions in Qualtrics. Typically, participants are given the choice to skip questions or not answer questions (and this is explained to them in the informed consent process). While it does increase missing data, it is important to give participants the option to skip questions they wish to skip so they do not feel coerced into participation, especially when being paid.

The authors should share more about their analysis plan for the qualitative analyses. How will they know when saturation has occurred for given themes? Will one researcher do all of the coding? If multiple coders, what will the consensus process be?

Reviewers' comments:

Reviewer's Responses to Questions

**Comments to the Author**

1. Does the manuscript provide a valid rationale for the proposed study, with clearly identified and justified research questions?

Reviewer #1: Yes

Reviewer #2: Yes

2. Is the protocol technically sound and planned in a manner that will lead to a meaningful outcome and allow testing the stated hypotheses?

Reviewer #1: Yes

Reviewer #2: Partly

3. Is the methodology feasible and described in sufficient detail to allow the work to be replicable?

Reviewer #1: Yes

Reviewer #2: Yes

4. Have the authors described where all data underlying the findings will be made available when the study is complete?

Reviewer #1: Yes

Reviewer #2: Yes

5. Is the manuscript presented in an intelligible fashion and written in standard English?

Reviewer #1: Yes

Reviewer #2: Yes

You may also provide optional suggestions and comments to authors that they might find helpful in planning their study.

Reviewer #1: Summary:

The present article reports an original clinical trial protocol to test Thriving with Bipolar Disorder, a peer-led psychoeducational group program developed with a community-based participatory (CBPR) framework. Specifically, the authors describe the planned methods for evaluating the feasibility of this single-arm pilot clinical trial in a series of four groups of approximately 10 participants, while collecting fidelity, acceptability, and outcome data. They also proposed to collect qualitative data from a subset of participants and peer facilitators through feedback interviews. Strengths include focusing on an important topic of developing the peer workforce and enhancing bipolar disorder recovery, previously registering the protocol with clinicaltrials.gov, including a detailed Table 1 description of session topics and objectives, inclusion of the Behavioural Research Ethics board-approved protocol, and partnering with the Hope+Me charity to deliver the intervention. However, the selected SPIRIT checklist version (2013 instead of 2025), minor points about the intervention, and self-rated fidelity assessment temper the enthusiasm for this manuscript, although these points are addressable.

Major:

N/A

Minor:

Abstract

1. N/A

Introduction

1. N/A

Intervention

1. Pages 8-9: Given the high rates of substance use among individuals with bipolar disorder, is substance use discussed in one of the self-management session topics (e.g., physical health)? If not, it might be worth considering adding or assessing in the feedback interview which topics, if any, participants would want to add to the program.

2. Page 10: It may be helpful to update the expected study dates (e.g., recruitment of program participants is described as concluding by August 2025, but as of September 12, 2025, the clinicaltrials.gov listing states that the study is not yet recruiting.

Procedures

1. Pages 14-15: Facilitator-reported fidelity presents challenges in that people tend to rate their own performance more favorably than objective fidelity raters. Although having an external observer is noted as potentially impacting participant willingness to participate in group discussions and activities, the option to unobtrusively record sessions, as is easily facilitated on Zoom, is not discussed and could be a viable option to rate fidelity. Given the pilot nature of this project, this would be a good opportunity to at least ask participants whether they would be open to recording to ensure that the facilitators are conducting the group in the best way possible or similar rationale, which will provide data to inform how fidelity would be monitored in future randomized controlled trials.

Discussion

1. N/A

Supplemental

1. The authors include the 2013 SPIRIT checklist. Given the availability of the 2025 SPIRIT checklist, it seems most appropriate to use the current checklist: https://www.consort-spirit.org/.

Reviewer #2: This manuscript describes the protocol for a single-arm pilot study designed to evaluate the feasibility (main outcome: session attendance), fidelity, acceptability, short-term participant impact and perceptions of a peer-delivered, group psychoeducational program for persons living with bipolar disorder. The manuscript is well written and the methodology detailed in it is generally sound. My main comments/suggestions are related to the data analysis.

Major comments:

- Analysis of the main outcome: Please include a description of the statistical methods that will be used to analyse the main outcome (session attendance). Will any inferential procedures be conducted? As session attendance will be measured only once per participant (i.e., number of sessions attended), repeated measures ANOVA is not appropriate as an analytic method (page 19, lines 425-426). Also, the per-protocol and intention-to-treat analyses are not relevant, as all individuals will have all relevant data for this outcome.

- Analysis of other outcomes: Although repeated measures ANOVA is appropriate for other outcomes that may change over time (and which are, thus, clustered within the subject), it has several problems relative to modern longitudinal methods, most notably (i) restrictive assumptions (sphericity is unlikely when an intervention has an effect) and (ii) inflexibility to handle missing data (see comment below). Moreover, repeated measures ANOVA cannot handle multiple levels of clustering as will occur in the proposed research (clustering of longitudinal measures within subject and also clustering of subjects within facilitator/peer group). Simple alternatives would include either using (a) linear mixed models with REML (due to the small sample size) or (b) repeated measures ANOVA supplemented by (i) tests for sphericity and adjustments, (ii) modern methods for assessing and handing missing data (see comment below) and (iii) adjusting for facilitator/peer group as fixed effects (acknowledging limitations from having a small sample size).

- Assessment of missing data: I would suggest the authors to describe the methods that will be used to examine the mechanisms and correlates of missingness, as well as informing the handling of missing data (see comment below)

- Handling of missing data: The authors propose to handle missing data by using simple imputation (last observation carried forward) in intention-to-treat analyses (page 19, lines 434-435). However, this method is suboptimal relative to more modern alternatives and has a high risk of introducing bias even in the least stringent scenarios. I would suggest the authors to (a) use models that can handle missing data more flexibly under less restrictive scenarios (e.g., linear mixed models) or (b) use modern missing data methods. Optionally, the authors may find it useful to implement sensitivity analyses assuming "worst-case" scenarios for missing data.

Minor comments:

- Qualitative interviews: Please clarify if sampling for the in-depth interviews will be based on any participants' characteristics (and the criteria that will be used).

-- Typographical errors:

- Page 17, lines 364-365: Please correct the phrase “(…) where formal psychiatric evaluation and support is not be available (…)”

- Please add page numbers for reference 4

- Please correct the formatting of page numbers in references 5, 39, 46 and 64

- Please replace the DOI with the volume, issue and page numbers for reference 11

- Please correct the page numbers in reference 19

-Please add the issue number for reference 27

- Please quote the supplement number inside brackets in reference 60

-Please add the volume, issue and page numbers for reference 70

**Do you want your identity to be public for this peer review?**  For information about this choice, including consent withdrawal, please see our Privacy Policy

Reviewer #1: No

Reviewer #2: No

---

## [Author Response · Author response to Decision Letter 1]

2 Nov 2025

A copy of our response to reviewers has been uploaded as a file in this resubmission - see "TWBD Protocol - Response to Reviewers.docx". While we have also included these comments here, we feel these are easier to read and follow in the formatted letter.

Editor Comment: Please ensure that your manuscript meets PLOS ONE's style requirements, including those for file naming.

Response: We can confirm we have reviewed the linked style templates to ensure our manuscript meets PLOS ONE’s style requirements.

Editor Comment: Thank you for stating the following in the Competing Interests section:

“EM has received honorarium for advising on the development of unrelated educational materials for Neurotorium, an online educational platform supported by the Lundbeck Foundation. AK declared no competing interests with respect to the research, authorship, and/or publication of this article. EEM has received funding to support patient education initiatives from Otsuka-Lundbeck.”

Response: We have updated the competing interests statement in the cover letter.

Editor Comment: Please include captions for your Supporting Information files at the end of your manuscript, and update any in-text citations to match accordingly.

Response: Thank you for flagging this. We have updated the captions accordingly.

Editor Comment: Recommend rewording the first sentence to say “BD is a serious, potentially chronic mental health condition” – as symptoms are significant, no matter their duration.

Response: Thank you for this feedback, we have updated the sentence accordingly.

Editor Comment: The authors cite a meta-analysis of 17 peer-facilitated self-management groups for those with smi – could they specify how many of those participants across studies these studies had a diagnosis of BD? This would help establish that the uniqueness of the present study.

Response: We appreciate this suggestion, and have elaborated on the studies included in this systematic review and their inclusion of / tailoring to individuals with BD. This text now reads: “A meta-analysis of 17 peer-facilitated self-management and recovery education interventions for individuals with severe and long-term mental health conditions demonstrated small-to-medium effects for symptom severity, self-perceived recovery, hopefulness, and empowerment (22). Only one program included in this review specifically focused on BD, however in this self-management education program peers were involved as coaches and did not deliver the psychoeducational material themselves (21). Of the other included studies, 12 included people with BD and two included individuals with unspecified affective disorders as participants in program evaluations open to individuals of varying diagnoses, however program content was not specifically tailored to BD.”

Editor Comment: For the qualitative analyses, it is unlikely saturation will be reached if only interviewing four facilitators; consider increasing this sample if possible.

Response: We agree that this is a small sample. However, as only four facilitators are being trained in the intervention for the protocol, we will not be able to increase the qualitative sample further. We have added a comment in the limitations section to note that there is a small qualitative sample and the potential impacts of this: “In addition, the facilitator qualitative interview sample size will be constrained by the number of facilitators trained in the intervention (n=4). Although the goal of qualitative research is not generalizability (84), a small interview sample may limit the depth and diversity of facilitator experiences explored. However, we note that such pragmatic considerations are an accepted justification for constraining sample sizes in published qualitative research (85).”

Editor Comment: Lots of detail is provided about the quantitative measures that will be given, but the qualitative interview guide is not as specific. Consider including questions that will be asked to parallel the specificity of quantitative measures described.

Response: Thank you for highlighting that additional information on the qualitative interview would be beneficial. To support this, we have provided a copy of both facilitator and attendee semi-structured interview guides as a supplementary file.

Editor Comment: The authors state that participants will not be able to skip questions in Qualtrics. Typically, participants are given the choice to skip questions or not answer questions (and this is explained to them in the informed consent process). While it does increase missing data, it is important to give participants the option to skip questions they wish to skip so they do not feel coerced into participation, especially when being paid.

Response: Thank you for sharing your perspective on the ability to skip questions in Qualtrics. Although our ethics review board provided approval for the study to proceed without the ability to skip items, we acknowledge that some participants may prefer the option to skip questions they do not wish to answer. Given that data collection is currently underway we are not able to make this suggested modification at this point in time, however it is certainly an option we will discuss with CREST.BD advisory groups in preparation for any future randomized control trial.

Editor Comment: The authors should share more about their analysis plan for the qualitative analyses. How will they know when saturation has occurred for given themes? Will one researcher do all of the coding? If multiple coders, what will the consensus process be?

Response: To address this comment, we have added some additional information about the plan for qualitative data analysis. We note that we will be following recommendations for reflexive thematic analysis; as such, only one analyst will perform the coding.

This is now elaborated on in text: “Qualitative data analysis will be performed using Braun and Clark’s guidelines for reflexive thematic analysis (74) using NVivo 14 (QSR International, 2023). Data familiarization will occur through reading and re-reading transcripts. Inductive coding approach will be used to assign brief descriptive codes derived from the data. Themes will be generated to describe patterns in coding. Provisional themes will be reviewed during development for coherency, meaningfulness, richness, and relationships to other themes. Results will be presented as a narrative summary of the most salient and relevant themes with respect to the study aims.

In line with quality practice recommendations for reflexive thematic analysis, a single researcher will perform the analysis (75). The primary analyst will maintain a reflexive journal to document the potential influence of any preexisting knowledge, assumptions, or personal characteristics on the data collection and analysis process. To support rigor, credibility will be established via peer debriefing between the co-authors EM and EEM to reflect on key decisions during data collection and analysis (76). Credibility will also be upheld through use of member checking, which will be accomplished by inviting CREST.BD advisory group members to review and comment on the definition, relevance, and interpretation of themes developed (77). Confirmability of the results will be supported through the use of verbatim participant quotes.”

Our approach to sample size determination was guided by the developers of reflexive thematic analysis, who recommend against the use of saturation (for a detailed comment, please see Braun and Clarke’s 2021 paper, “To saturate or not to saturate”). In brief, failure to generate new themes (a criteria used to determine saturation) does not safeguard against thin, poorly detailed, or irrelevant responses: technically, saturation could be quickly reached by interviewing a sample with limited experience or interest in the topic, or by using poor interviewing techniques (i.e., closed questions, limited probing). Furthermore, saturation has been critiqued for implying a positivist perspective on data (i.e., that there are a limited amount of themes). In contrast, Braun and Clarke position that decisions about when to stop collecting data are inherently subjective (and should therefore be subject to the same type of reflexivity and rigour as other analytic decisions undertaken using reflexive thematic analysis). Information power is an approach to determining sample size that is congruent with reflexive thematic analysis, where the richness and relevance of data obtained to date are reflexively considered when making a determination about when to stop collecting data.

We had previously noted in the data collection section that information power would be used; the rationale for this choice has now been elaborated on with the following text: “As saturation is not a recommended approach for reflexive thematic analysis (65), the initial target sample size for the qualitative interviews will be guided by both information power (66) and feasibility considerations (i.e., the number of facilitators interviewed will be constrained by the number of trained facilitators, and the number of attendees interviewed will be constrained by the number of participants who opt-in to being contacted). Aspects of information power that support the use of a smaller initial target sample were the narrowness of the study aim (that is, to evaluate the feasibility, acceptability, and preliminary outcomes of this program), specificity of the sample (program delivery will be manualized), and expected quality of the dialogue (EM is an experienced qualitative interviewer with expertise in peer support and BD self-management interventions). Information power is intended to be a flexible approach to sampling, and as such the appropriateness of the attendee qualitative interview sample size will be iteratively reviewed as data collection progresses. The richness and relevance of the interview dialogue will be discussed between research team members to inform the final sample size.”

Reviewer 1 Comment: The present article reports an original clinical trial protocol to test Thriving with Bipolar Disorder, a peer-led psychoeducational group program developed with a community-based participatory (CBPR) framework. Specifically, the authors describe the planned methods for evaluating the feasibility of this single-arm pilot clinical trial in a series of four groups of approximately 10 participants, while collecting fidelity, acceptability, and outcome data. They also proposed to collect qualitative data from a subset of participants and peer facilitators through feedback interviews. Strengths include focusing on an important topic of developing the peer workforce and enhancing bipolar disorder recovery, previously registering the protocol with clinicaltrials.gov, including a detailed Table 1 description of session topics and objectives, inclusion of the Behavioural Research Ethics board-approved protocol, and partnering with the Hope+Me charity to deliver the intervention. However, the selected SPIRIT checklist version (2013 instead of 2025), minor points about the intervention, and self-rated fidelity assessment temper the enthusiasm for this manuscript, although these points are addressable.

Response: We appreciate this positive feedback on the strengths of the protocol. We hope to address the points raised by the reviewer below.

Reviewer 1 Comment: Pages 8-9: Given the high rates of substance use among individuals with bipolar disorder, is substance use discussed in one of the self-management session topics (e.g., physical health)? If not, it might be worth considering adding or assessing in the feedback interview which topics, if any, participants would want to add to the program.

Response: Thank you for this suggestion. We agree that substance use is an important aspect of physical health, and this is reflected in the quality of life framework that this program, and the preceding interventions that informed it (i.e., the Bipolar Wellness Centre, PolarUs), are based on. In this framework, physical health incorporates diet, exercise, sexual health, and substance use. However, for feasibility reasons, we are not able to cover all 17 aspects of quality of life in the context of this group program. In order to determine the quality of life topics and strategies for inclusion in this program, we consulted with CREST.BD community advisory groups as described in Phase 1 of the protocol. We note that substance use was flagged as a potentially sensitive topic that may be potentially embarrassing or triggering to discuss in a group setting, and may not be relevant to all participants, with some CREST.BD advisory group members strongly discouraging its inclusion. For this reason, substance use was not selected as one of the focus topics for the group program.

However, we acknowledge that the perspectives of the community advisory groups and program participants may differ. Related to this reviewer’s comment, we hope to obtain feedback on program content that participants would like to see added or changed. All qualitative interview participants will be asked “If you could change or add anything to this program, what would it be?” Additional probes will be used to explore perceptions of program content, delivery, the facilitator, and the group environment. We have added a sentence to the description of the qualitative interview in the Data Collection section to emphasise this: “Participants will also be asked for their perspectives on aspects of the program that should be modified or added.” We have also added a sentence to the discussion, noting how this knowledge will be used to iterate the program: “Such insights will be used to further refine the program content, delivery, training materials, and evaluation methods for future trials.” The qualitative interview guides have been included in this revision as a supporting file (S3).

Reviewer 1 Comment: Page 10: It may be helpful to update the expected study dates (e.g., recruitment of program participants is described as concluding by August 2025, but as of September 12, 2025, the clinicaltrials.gov listing states that the study is not yet recruiting.

Response: Thank you for flagging this. The expected study dates listed in the protocol manuscript are correct; we have updated the clinicaltrials.gov listing.

Reviewer 1 Comment: Pages 14-15: Facilitator-reported fidelity presents challenges in that people tend to rate their own performance more favorably than objective fidelity raters. Although having an external observer is noted as potentially impacting participant willingness to participate in group discussions and activities, the option to unobtrusively record sessions, as is easily facilitated on Zoom, is not discussed and could be a viable option to rate fidelity. Given the pilot nature of this project, this would be a good opportunity to at least ask participants whether they would be open to recording to ensure that the facilitators are conducting the group in the best way possible or similar rationale, which will provide data to inform how fidelity would be monitored in future randomized controlled trials.

Response: We appreciate this point of feedback, and acknowledge that there is a tension between data collection methods that could ensure a more comprehensive assessment of fidelity to the manual, and the potential of observation or recording to impact attendee’s comfort and willingness to engage in the program. We have added additional discussion of this issue to the Limitations section to make our rationale for self-repor

---

## [Decision Letter · Decision Letter 1]

20 Nov 2025

‘Thriving with bipolar disorder’: The co-design of a peer-delivered group psychoeducation program and single-arm pilot feasibility evaluation protocol.

PONE-D-25-37425R1

Dear Dr. Michalak,

We’re pleased to inform you that your manuscript has been judged scientifically suitable for publication and will be formally accepted for publication once it meets all outstanding technical requirements.

Kind regards,

Rakesh Karmacharya, MD, PhD

Academic Editor

PLOS ONE

Additional Editor Comments (optional):

Reviewers' comments:

Reviewer's Responses to Questions

**Comments to the Author**

1. Does the manuscript provide a valid rationale for the proposed study, with clearly identified and justified research questions?

Reviewer #2: Yes

2. Is the protocol technically sound and planned in a manner that will lead to a meaningful outcome and allow testing the stated hypotheses?

Reviewer #2: Yes

3. Is the methodology feasible and described in sufficient detail to allow the work to be replicable?

Reviewer #2: Yes

4. Have the authors described where all data underlying the findings will be made available when the study is complete?

Reviewer #2: Yes

5. Is the manuscript presented in an intelligible fashion and written in standard English?

Reviewer #2: Yes

You may also provide optional suggestions and comments to authors that they might find helpful in planning their study.

Reviewer #2: The authors have addressed all of my comments thoroughly and satisfactorily. I have no further concerns about the proposed statistical analysis and the reporting of the study protocol.

**Do you want your identity to be public for this peer review?** For information about this choice, including consent withdrawal, please see our Privacy Policy

Reviewer #2: No

---

## [Editor Report · Acceptance letter]

PONE-D-25-37425R1

PLOS One

Dear Dr. Michalak,

I'm pleased to inform you that your manuscript has been deemed suitable for publication in PLOS One. Congratulations! Your manuscript is now being handed over to our production team.

Kind regards,

on behalf of

Professor Rakesh Karmacharya

Academic Editor

PLOS One